# Can Dog-Assisted Intervention Decrease Anxiety Level and Autonomic Agitation in Patients with Anxiety Disorders?

**DOI:** 10.3390/jcm10215171

**Published:** 2021-11-04

**Authors:** Dorota Wołyńczyk-Gmaj, Aleksandra Ziółkowska, Piotr Rogala, Dawid Ścigała, Ludwik Bryła, Bartłomiej Gmaj, Marcin Wojnar

**Affiliations:** 1Department of Psychiatry, Medical University of Warsaw, 00-665 Warsaw, Poland; Dorkens@gmail.com (D.W.-G.); aleksandra-ziolkowska@wp.pl (A.Z.); rogala.piotr91@gmail.com (P.R.); marcin.wojnar@wum.edu.pl (M.W.); 2Institute of Psychology, The Maria Grzegorzewska University, 02-353 Warsaw, Poland; dscigala@aps.edu.pl; 3Oddział XIII Dzienny Zaburzeń Nerwicowych Szpitala Nowowiejskiego, 00-773 Warsaw, Poland; ludwik.bryla@gmail.com

**Keywords:** animal assisted intervention (AAI), dogs, dogotheraphy, anxiety, anxiety disorders treatment, heart rate

## Abstract

Few studies have explored the influence of an Animal-Assisted Intervention on patients with mental disorders. We investigated it’s impact on anxiety symptoms. We divided 51 patients with anxiety symptoms into two groups—treatment group, that went for a short 15–20 min’ walk with a dog, his handler and a researcher and control group, that went for a walk only with a researcher. We used State-Trait Anxiety Inventory (STAI), Visual Analogue Scale (VAS) of fear, Beck Depression Inventory (BDI), Ford Insomnia Response to Stress (FIRST), Brief symptom Inventory (BSI) and VAS of satisfaction after trial to assess. We also checked the resting blood pressure and resting heart rate before and after performing psychological tests while sitting. We have obtained full data of 21 people from the research group and 26 people from the control group. After the intervention, the treatment group reported lower anxiety levels as a state (Mean (M) = 34.35; Standard Deviation (SD) = 6.9 vs. M = 40.94; SD = 8.6) and fear (M = 1.05; SD = 1.0 vs. M = 2.04; SD = 2.2) than the control group. After a walk with a dog, trait anxiety (M = 34.35; SD = 6.9 vs. M = 46.3; SD = 9.6), state anxiety (M = 48.9; SD = 7.2 vs. M = 53.9; SD = 7.8), fear (M = 1.05; SD= 1.0 vs. M = 2.57; SD = 2.3) and resting heart rate (M = 71.05; SD = 12.3 vs. M = 73.67; SD = 13.1) decreased significantly, while walking without a dog only reduced state anxiety (M = 47.24; SD = 11.0 vs. M = 40.94; SD = 8.6). Multivariate analysis of variance showed that after the walk, state anxiety was significantly lower in the treatment group than in the control group, F(1.35) = 6.706, *p* <0.05, *η^2^* = 0.161. Among those who walked with a dog, the intervention also led to significant decreases in fear and resting heart rate, F(1.44) = 11.694, *p* < 0.01, *η^2^* = 0.210 and F(1.45) = 8.503; *p <* 0.01; *η^2^* = 0.159, respectively. For anxious patients, a short walk with a dog is more beneficial than a walk without one. We found significant positive effects of a dog’s company on vegetative arousal and mental comfort. This is another study confirming the possible therapeutic effect of the animal on anxiety symptoms. Further research is required, especially in the large groups of patients, as recommendations on the use of Animal Assisted Interventions (AAI) are needed.

## 1. Introduction

Dogs are “man’s best friend”, this popular phrase largely reflects the increasing scientific knowledge of the interaction between the two species. Anthropology, archaeology and genetics provide ample evidence for a long-lasting, very close and mutually beneficial relationship between people and dogs. Medical sciences such as immunology or endocrinology also found the possible positive influence of a dog’s presence on humans. Finally, mental health science is increasingly acknowledging the dogs’ therapeutic effect on many mental disorders [1,2,3].

Anxiety is a natural physiological reaction to threat, but if it is excessive, inadequate, non-adaptive or impairing one’s performance, it can be a symptom of numerous mental illnesses, especially depression and anxiety disorders. One of the mental disorders most common are anxiety disorders, their annual prevalence in Europe is 14% [4] They entail emotional, somatic and cognitive symptoms, as well as behavioral disturbances, difficulties in everyday life especially in the social interactions. Anxiety disorders increase the risk of somatic diseases, e.g., cardiovascular diseases and deteriorate the quality of life [5,6]. The cause-and-effect relationships between anxiety disorders and somatic diseases are reciprocal [7], e.g., fear of surgical treatment may make it difficult or even impossible to properly treat many somatic diseases, but also trigger anxiety disorders [8].

The treatment of anxiety disorders is currently based on pharmacotherapy and different forms of psychotherapy, especially cognitive behavioral therapy—CBT [9] but many other forms of therapy are also being tested. Alternative methods of treatment are needed, because the effectiveness of the currently recommended treatment of anxiety disorders is still not satisfactory, whereas anxiety disorders are a serious social problem due to their high prevalence and health consequences [4,10]. It seems that the support of specifically trained assistance dogs is a promising supplement to existing anxiety therapies [11,12].

In general, any intervention that intentionally involves animal as part of a therapeutic process falls under the realm of Animal-Assisted Interventions (AAI) [1]. In psychiatry especially Animal-Assisted Therapy (AAT) and Animal-Assisted Activity (AAA) are tested. AAT are the professional, goal-directed therapeutic interventions with animals specifically trained for that purpose and are delivered by trained professionals, working formally and directly with healthcare, educators or/and human providers. Animal-Assisted Therapy, especially using horses and dogs, is becoming increasingly common, especially in the treatment of patients with autism spectrum disorders, dementia and Post Traumatic Stress Disorders (PTSD). AAA means interventions that are less structured interactions with animals suitable for this purpose, that have positive impact on the quality of life but are not necessarily delivered by trained professionals. Apart from the aforementioned animal interventions forms, patients can also receive a dog trained to assist them in daily life. Such dogs can, for instance, bring the patient medicine or remind them to take it, fetch a phone in case of panic attack or help find their way back home. There are three types of assistance dogs: guide dogs, hearing dogs, and service dogs. While the guide dogs are recruited from larger breeds, especially Labrador Retriever, hearing dogs and service dogs especially emotional support dogs may also be of small as Chihuahua [13].

As studies have shown, the presence of a dog may have a positive influence on mood, health and the quality of life. It can reduce the level of anxiety and enhances positive emotions, as it is associated with an increased secretion of oxytocin to plasma [2,3]. Even short (no longer than 30 min) AAI sessions reduce the level of anxiety in patients with mood disorders, psychotic disorders or drug addictions [1]. They can also decrease the anxiety-related arousal in patients waiting for electroconvulsive therapy (ECT) as well as pain in patients with fibromyalgia [14,15]. Assistance dogs can make patients with anxiety disorders feel safer, they facilitate social activity and reduce the feelings of isolation and loneliness [16,17,18,19]. Previous studies confirmed that AAI may lead to short-term improvements in depression, anxiety and PTSD symptoms [20]. In case of PTSD patients, the presence of a dog may also remind them that the threat has already passed. On top of that, systematic meetings with a dog can significantly reduce the symptoms of anxiety and increase mental comfort in a health care setting [21]. However, it should be emphasized that not all patients may benefit from contact with an assistance dog—dog hair allergy and dogophobia are the main contraindications. There is still little research on the effectiveness of AAI in the group of people with anxiety disorders. Because loneliness and lack of support affects negatively the course of anxiety disorder, it seems that this particular group could especially benefit from the animal assistance [10]. As far as we know, our study is the first AAA study in Poland. Studies demonstrating the potential effectiveness of AAI increase the probability of creating legal regulations, as well as recommendations for this alternative treatment, which can contribute to improvement of the quality of life of patients with anxiety disorders.

The purpose of our research was to investigate whether interactions with the service dog can reduce the anxiety symptoms in the patients with anxiety or mixed depressive-anxiety disorders. We were assessing the impact of a short (15–20 min duration) animal assisted intervention- a one-time walk of a patient with anxiety disorders or mixed anxiety-depression adjustment disorders (ICD-10: F40-43) with a trained dog, his handler and a doctor. The control group also had one-time walk but just with a doctor. Based on previous studies, we have assumed that patients who walk with a trained dog compared to those who walk without a dog, will have lower anxiety scores and lower resting cardiovascular parameters such as resting pulse and resting blood pressure.

## 2. Materials and Methods

The research was conducted in 2015 and 2016 at the Nowowiejska Hospital (Warsaw, Poland) day ward and in its small garden. All participants of our study were recruited from patients treated in this ward with group psychotherapy in little subgroups, that were finishing their therapy between noon and 5 pm. We included to our study only patients with clinical diagnosis of anxiety or mixed depressive-anxiety disorders according to ICD10 (F40-F43.2). Patients were informed about our study by their therapists, then the volunteers were explained the details of the study by the researchers. All of them completed the same questionnaires before and after the interventions. All interventions were performed at the same time of the day (between noon and 5 pm), immediately after daily group psychotherapy when all other patients were free to go home. We examined one participant at a time, and, considering the welfare of the service dog, usually 1–3 patients per day (immediately after finishing their therapy). This day ward was well suited to conduct our study, because we had an access to many patients with anxiety disorders. Moreover green and calm surroundings of the ward allow for comfortable and undisturbed contact of subjects with the serving dog. The dog used in this research was a therapy dog—2.5 years old, female German shepherd. The owner of the dog is a man, who established a Service Dogs Poland foundation, dedicated to training service dogs and promoting the beneficial contact between dogs and humans. This dog was used to work with his handler in public places such as schools and kindergartens and to contact with many people interested in it and seemed to like it. It was provided with social play and rest to avoid overload. The study did not affect the quality of psychotherapy. Patients were provided with comfortable conditions during the study and they could stop it at any time. We chose such a study design because we wanted to have an access to patients with symptomatic anxiety disorders. Patients had contact with dog outside the ward because, due to statute of the hospital, dog wasn’t allowed to enter this building. The Warsaw Medical University Ethical Committee for scientific research approved the study and all participants provided informed consent to participate.

### 2.1. Participants

Participants in this study were volunteer patients with anxiety or adjustment disorder (based on ICD-10: F40-43) recruited during group therapy courses in the day treatment ward mentioned above. They were between 18 and 60 years old (M = 33.87 ± 10.410, 77% female). Exclusion criteria were: dogophobia, allergy to dog hair and a bad somatic condition, but none of the volunteers met it. 

Participants were randomly assigned to one of two conditions: the treatment condition in which they took a 15–20 min long walk with the assistance dog and its handler or the control condition in which they went for a walk without the dog, only with a medicine student or a doctor. Primarily the treatment group consisted of 25 and control group of 26 patients, but finally we obtained the full data from 21 patients from the treatment group (17 female and 4 male) and 26 patients (19 female and 7 male) from the control group. The two groups did not differ in terms of demographics or any of the measured psychometric and physiological variables. 

### 2.2. Instruments

#### 2.2.1. Questionnaires

All participants completed a questionnaire consisting of:A socio-demographic survey including questions about age, gender, marital status, education level, somatic disorders, pregnancy, animal allergies, having a dog, attitude to animals and subjective evaluation of own health—fulfilled directly before the intervention;The Beck Depression Inventory (BDI)—21-item self-report scale, each item consists of responses graded from 0 to 3, to assess the severity of depressive symptoms. [22,23] Cronbach’s alpha for our participants was 0.889; performed directly before and after the intervention;The Ford Insomnia Response to Stress Test (FIRST)— a self-rating questionnaire measuring the likelihood of the occurrence of sleep disturbances in response to commonly experienced stressful situations, consists of nine items with evaluation according to a 4-point Likert (1, not likely; 4, very likely), with total score 9–36 [24] Cronbach’s alpha in our study group was 0.859. It was performed directly before and after the intervention;The Brief Symptom Inventory (BSI-18)- a short version of the Symptom-Checklist (SCL-90-R), measures psychiatric symptoms and global symptom load. The BSI-18 comprises 18 items assessing psychological distress in the last seven days on three subscales (Depression, Anxiety, and Somatization) [25]. The Cronbach’s alpha for this scale for our participants was 0.935. It was performed directly before and after the intervention;The visual analogue scale (VAS fear) of level of fear—performed directly before and after the intervention;The State-Trait Anxiety Inventory (STAI)—a questionnaire measuring the level of anxiety and its changes. It consists of 40 items, half of which assess state anxiety, and the other half measure trait anxiety. Participants rate each item on a 4-point scale, ranging from “1 = not at all” to “4 = totally agree” in case of state anxiety, and “1 = almost never” to “4 = almost always” in case of trait anxiety. Participants’ responses are summarized, so for each of the two types of anxiety the results range from 20 (very low anxiety level) to 80 (very high anxiety level) [26]. The reliability of the scales in our study was, respectively, for state α = 0.932 and trait α = 0.875—performed directly before and after the intervention;The visual analogue scale (VAS satisfaction) of level of satisfaction with the intervention—performed directly after the intervention.

The visual analogue scales were 11-point scales ranging from 0 = ‘no fear’ in case of the VAS fear scale and ‘completely dissatisfied’ in case of the VAS satisfaction scale, to 10 = ‘strongest possible fear’ and ‘completely satisfied’, respectively.

#### 2.2.2. Physiological Measures

We measured resting heart rate, resting systolic blood pressure and resting diastolic blood pressure, because they are related to the mental state. The measurements were taken twice—first time directly after the first psychological assessment with scales and second time, after the second psychological testing. During performing tests, the patients were sitting for about 15 min, so walking movement should not affect the resting cardiovascular results. We always used the same device, with the cuff placed around the right upper arm of the participant.

### 2.3. Interventions/Procedures

#### 2.3.1. Animal-Assisted Intervention Group (AAI Group)

In the treatment condition, a participant first completed the first part of the questionnaires. Then, a medicine student or doctor measured his/her resting pulse and resting blood pressure. Directly afterwards, the participant exited the building and was introduced to the dog’s handler and the assistance dog, who were already waiting in the surrounding green area. The subject hasn’t been exposed to this dog before. The handler was asked to focus the conversation on the dog and the participant’s experience with dogs in general and he could also show the skills of his dog, so the test conditions were comparable for all subjects. Physical contact with the dog (such as petting) was permitted, the participant was free to decide on the level of interaction. All subjects were mostly slowly walking, but they could also stop for a while. After 15–20 min with the dog, the participant went back to the building and completed the second part of the questionnaires. As soon as he/she finished (10–30 min after the intervention) we took resting pulse and resting blood pressure measurements again.

#### 2.3.2. Control Group

In the control condition, the experimental procedure was analogous, except that the walk was just with a medicine student or a doctor, without the dog and its handler. The student / doctor was asked to focus the conversation on the study, the assistance dog and the participant’s experience with pets in general. The control subjects have never been exposed to this dog. All other procedures were exactly the same as in the treatment group.

## 3. Analysis and Results

We used SPSS 25 (International Business Machines Corporation (IBM), NY, USA) to run all the analyses. At the beginning, the basic parameters of the analyzed variables (M; SD) were analyzed (Table 1). The analysis of the results of the measures of skewness and kurtosis was carried out to determine whether they fall within the range from −1 to +1, which allows to confirm the normal distribution of the analyzed variables. Then we independently compared the treatment (walk with a dog) and the control (walk without a dog) groups before (1st trial) and after (2nd trial) the intervention. Then we conducted mixed analyses of variance on the variables most affected by our intervention by using one-way analysis of variance. The next step was to conduct a multivariate analysis of variance, including the comparison of changes in psychological (anxiety) and physiological parameters (the resting blood pressure and heart rate) within each group, as well as presentation and interpretation of the effect of interaction of the measurement factor over time (measurement 1 vs. measurement 2) with the factor group (research vs. control). The first step of MANOVA was to verify the interaction effect of the test condition (walk with the dog vs. walk without a dog) with the level of anxiety as the state in measurement 1 and 2. The next step was to verify the effect of the interaction of the test condition with the level of fear in measurement 1 and 2. Finally, interaction of the test condition with the resting blood pressure of the subjects in measurement 1 and 2.

### 3.1. Between-Group Comparison before the Intervention

There were no significant differences between the two groups (treatment vs. control) with respect to age, F(1.45) = 0.001, *p* = 0.993 or gender distribution, χ^2^(1, N = 47) = 0.40, *p* = 0.53. The groups also did not differ in terms of anxiety as a state (STAI; M = 46.3; SD = 9.6 vs. 47.2 ± 11) F(1.35) = 0.076; *p* = 0.784, anxiety as a trait (STAI; M = 53.9; SD = 7.8 vs. M = 55.18; SD= 9.9) F(1.35) = 0.193; *p* = 0.663, sleep reactivity (FIRST; M = 24.58; SD = 6.7 vs. M = 26.79; SD = 6.1) F(1.34) = 1.046; *p* = 0.314, psychological distress (BDI; M = 22.00; SD = 8.1 vs. M = 19.69; SD = 10.9) F(1.43) = 0.607; *p* = 0.440 or fear (VAS fear; M = 2.57; SD = 2.3 vs. M = 2.48; SD = 2.55) F(1.44) = 0.016; *p* = 0.900. When it comes to physiological measures, we found no differences between the groups in resting systolic pressure (M = 125.05 mmHG; SD = 15.2 vs. M = 125.27 mmHG; SD = 17.3) F(1.45) = 0.002; *p* = 0.963, resting diastolic pressure (M = 76.33; SD = 13.1 mmHg vs. M = 74.42 mmHg; SD = 11.2) F(1.45) = 0.301; *p* = 0.586 or resting heart rate (M = 73.67; SD = 13.1 vs. M = 72.46; SD = 9.1); F(1.45) = 0.138; *p* = 0.712.

### 3.2. Between-Group Comparison after the Intervention

After the intervention, the treatment group reported significantly lower anxiety as a state (STAI; M = 34.35; SD = 6.9 vs. M = 40.94; SD = 8.6); F(1.35) = 6.706; *p* < 0.05; *η^2^* = 0.161 and fear (VAS fear; M = 1.05; SD = 01.0 vs. M = 2.04; SD = 2.2); F(1.45) = 3.776; *p* < 0.05; *η^2^* = 0.077 than the control group. Participants in the treatment condition were also more satisfied with the intervention (VAS satisfaction; M = 9.10; SD = 1.1 vs. M = 7.73; SD = 1.6; F(1,41) = 10.032, *p* < 0.01, *η^2^* = 0.197).

There were no significant differences between the two groups on any other measures, i.e., anxiety as a trait, sleep reactivity, psychological distress, resting systolic blood pressure, resting diastolic blood pressure or resting heart rate.

### 3.3. Within-Group Comparisons

The effect of the intervention (i.e., the difference between befor and after intervention) was more pronounced in the treatment condition than in the control condition. In the treatment condition, anxiety as a state (STAI; M = 34.35; SD = 6.9 vs. M = 46.3; SD = 9.6) F(1.19) = 26.640; *p* < 0.001; *η^2^* = 0.584, anxiety as a trait (STAI; M = 48.9; SD = 7.2 vs. M = 53.9; SD = 7.8) F(1.19) = 12.533; *p* < 0.05; *η^2^* = 0.397, fear (VAS fear; M = 1.05; SD = 1.0 vs. M = 2.57; SD = 2.3) F(1.20) = 12.005; *p* < 0.05; *η^2^* = 0.375 and heart rate (M = 71.05; SD = 12.3 vs. M = 73.67; SD =13.1) F(1.20) = 11.125; *p* < 0.05; *η^2^* = 0.357 decreased after the intervention. In the control condition, the only significant difference between the 1st and the 2nd trial was a decrease of anxiety as a state (STAI; M = 47.24; SD = 11.0 vs. M = 40.94; SD = 8.6) F(1.16) = 12.006; *p* < 0.05; *η^2^* = 0.429

### 3.4. Within and between-Group Comparisons

We first performed an analysis of variance in a mixed regimen (MANOVA) to verify the effect of the interaction between the treatment condition (walking with vs. without the dog; between-subjects factor) and the trial (1st or 2nd; within-subjects factor) on the level of anxiety as a state (STAI; see Figure 1).

The analysis revealed no significant differences between the two groups in the 1st trial, F(1.35) = 0.76; *p* = 0.784, but in the 2nd trial state anxiety in the treatment condition was significantly lower than in the control condition, F(1.35) = 6.706, *p* < 0.05, *η^2^* = 0.161.

The significant simple main effect of the trial was obtained. There was a decrease in the level of anxiety as a trait in measurement 2 in relation to measurement 1. These effect was significant both in case of the treatment condition, F(1.35) = 34.065, *p* < 0.001, *η^2^* = 0.493, as well as in the control condition, F(1.35) = 8.033, *p* < 0.01, *η^2^* = 0.187, albeit in the former group the difference was more pronounced.

We then performed an analysis of variance in a mixed regimen (MANOVA) to verify the effect of the interaction between the experimental condition (walking with vs. without the dog; between-subjects factor) and the trial (1st or 2nd; within-subjects factor) on the level of fear (VAS; see Figure 2).

The analysis yielded no significant differences between the two groups in fear in the 1st trial, F(1.44) = 0.16; *p* = 0.900), but in the 2nd trial the level of fear was slightly lower (tendency-level significance) in the treatment group than in the control group, F(1.44) = 3.162, *p* = 0.082, *η^2^* = 0.067. The simple effect of the trial on fear was significant in the treatment group, F(1.44) = 11.694, *p* < 0.01, *η^2^* = 0.210, but not in the control group, F(1.44).

Finally, we performed an analysis of variance in a mixed regimen (MANOVA) to verify the effect of the interaction between the experimental condition (walking with vs. without the dog; between-subjects factor) and the trial (1st or 2nd; within-subjects factor) on participants’ heart rate (see Figure 3).

An analysis of simple effects of the experimental condition revealed no significant differences between the two groups in neither the 1st trial nor in the 2nd trial. The simple effect of trial was significant in case of the treatment condition, F(1.45) = 8.503; *p* < 0.01; *η*^2^ = 0.159, but not significant in case of the control condition.

The obtained results partially confirm our assumptions—contact with the dog was correlated with a decrease in the level of anxiety and a decrease in heart rate, but no significant change in blood pressure was obtained.

## 4. Discussion

The aim of our study was to test the hypothesis that a contact with a dog could have a beneficial effect on anxiety symptoms in the patients with anxiety and mixed depressive-anxiety disorders. We investigated the level of anxiety and the resting cardiovascular parameters before and after the walk with a specially trained service dog and his handler in patients diagnosed with F40-F43. The control group consisted of patients diagnosed with F40-F43 as well, but walked without the dog nor his handler, only with a doctor or medical student. All subjects were assessed with scales such as STAI, BDI, FIRST, VAS of fear before and after the walk and VAS of satisfaction after the walk. After sitting and fulfilling the scales for about 10–30 min they have resting heart rate and resting blood pressure measured.

People with anxiety disorders feel anxious about the new situations, so the study itself could have resulted in a high score on the VAS fear and STAI state. However, after the walk, both STAI and VAS fear were lower in the research group than in the control group.

These results are consistent with our hypothesis and with existing literature showing a reduction of stress through interactions with animals [27,28,29]. Children with insecure attachment had a lower cortisol peak after a socially stressful situation when they were assisted by a real dog as compared to a dummy dog [30]. The presence of a real dog, just like that of a friendly human, reduced the feelings of anxiety during exposure to a traumatic film among healthy female volunteers [31]. In schizophrenia patients, the cortisol level in the saliva was lower during AAT intervention then during other therapeutic conditions [32]. The level of cortisol in the serum of children during venipuncture was also lower in the presence of a dog [27]. After an enjoyable interaction with the dog, the levels of oxytocin, endorphins and dopamine increase and the activity of hypothalamic-pituitary axis decreases [30]. The STAI scale has been already used before in the studies of the level of anxiety after interactions with the dog [11,12]. In these studies, similarly to our study, a drop in the STAI score was found after contact with the dog. A pilot study by Hoffman et al. (2009) carried out on twelve patients diagnosed with major depression (age: 40.5 ± 10 years) have shown a drop in the STAI state scores after short, 30 min contact with the animal. Similarly study by Giuliani et al. (2017) carried out on 53 adults (age: 36.5 ± 11.2 years) treated with psychotherapy for learning disabilities confirmed that the company of a dog during psychotherapy resulted in decline in the anxiety score measured with STAI state. Recent study show the advantage of psychotherapy along with the service dog over psychotherapy itself in veterans with posttraumatic stress disorder (PTSD) in term of depression and the satisfaction of life. The addition of trained service dogs to usual care may confer clinically meaningful improvements in PTSD symptomology [29]. 

Heart rate can be treated as an objective measure of the sympathetic system reactivity. Significantly lower resting heart rate of the patients who walked with a dog (vs without one), measured after sitting and completing the scales, allows us to conclude that the patients were objectively calmer after spending time with the animal. A significant reduction of systolic blood pressure, total cholesterol level and heart rate in the line with significant reduction of cortisol in dog companionship was proven in the review of cardiological studies [33].

However, despite there being some trials indicating that AAI could reduce not only heart rate, but also blood pressure [34,35], in our study we did not detect such a difference. We presume that was because physical activity and curiosity related to meeting the assistance dog were also factors affecting the participants’ blood pressure.

However there is still little number of publications about AAA and they involve different intervention types, procedures and designs [36]. Therefore their results are difficult to compare, draw conclusions and make recommendations. More research on this issue is needed.

We are aware of certain limitations of the described study, the most significant of them being the use of an assistance dog with its handler in the treatment condition. This way we did not differentiate between the effect of the handler and that of the dog itself. However, previous data shows that the patient response to a dog with his owner was greater than that to the owner alone [37]. Different breeds of dogs are used for AAI. Perhaps exposure to a small dog, the emotional dog, for some patients, could have a more pronounced anxiety-reducing effect. Another limitation is that the control group walked not only without a dog but also without an owner. So the study measured an effect of the walk with a team: the owner and the dog on the anxiety level and cardiac parametres. Moreover, the research group initially consisted of 25 people, decreased by 4 people due to incomplete data. We could not do additional research because of the departure of the dog’s handler. Thus the control group was larger. What is more, we cannot rule out that subjects took a significantly different number of steps while walking, which could also have influenced the results of the study. The well-being of the patients during the study could also be influenced by factors independent of the study, such as the weather. Another limitations is the small sample size thus this results may be incidental and unrepresentative. Finally, the weakness of our study is also that it was limited to self-report and not blinded.

Our study supported that a dog’s presence can have the beneficial influence on anxiety level in anxious/depressed patients. Our conclusion is that Animal Assisted Activity may be an effective adjunctive modality for the treatment of patients with anxiety and mixed depressive- anxiety disorders. The implementation of Animal Assisted Activities in the additional treatment of anxiety and depressive disorders should be considered. Further research is required, especially in the large groups of patients. Future studies should include additional objective neurophysiological measures and incorporate longitudinal testes into their design to assess the durability of such interactions. We would like to point out that the results of our study may contribute to increase of interest in Animal Assisted Activities as a support treatment of anxiety and depressive disorders in Poland. The psychiatric patients need new laws and regulations to protect their rights by allowing them reasonable access with their service dogs.

## Figures and Tables

**Figure 1 jcm-10-05171-f001:**
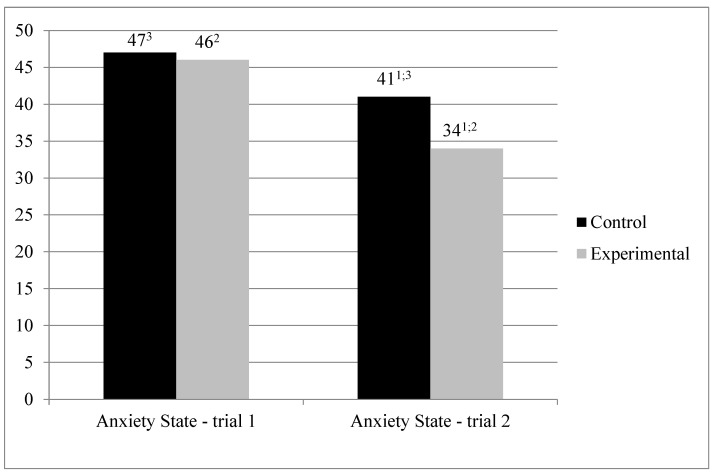
Average anxiety as a state (STAI). 1—in the 2nd trial state anxiety in the treatment condition was significantly lower than in the control condition (*p* < 0.05). 2—significant difference between trial 1 and 2 in treatment condition (*p* < 0.05). 3—significant difference between trial 1 and 2 in control condition (*p* < 0.05).

**Figure 2 jcm-10-05171-f002:**
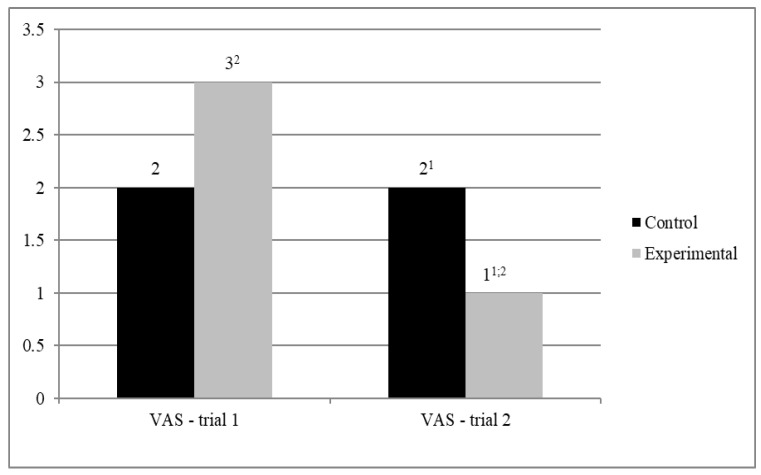
Average fear (VAS). 1—in the 2nd trial fear in the treatment condition was significantly lower than in the control condition (*p* < 0.05). 2—significant difference between trial 1 and 2 in treatment condition (*p* < 0.05).

**Figure 3 jcm-10-05171-f003:**
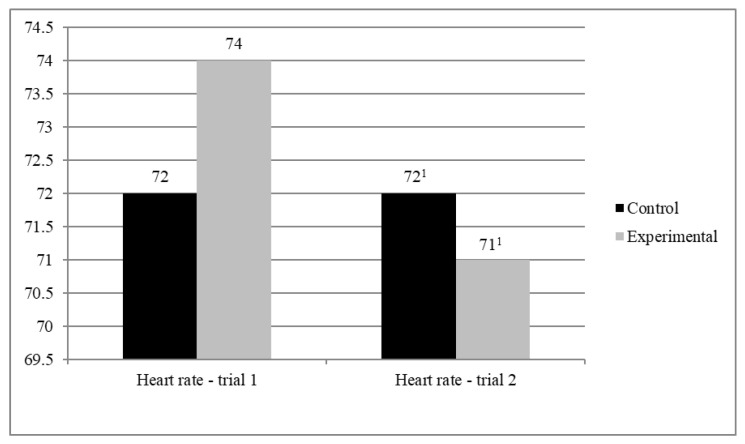
Average heart rate. 1—in the 2nd trial heart rate in the treatment condition was significantly lower than in the control condition (*p* < 0.05).

**Table 1 jcm-10-05171-t001:** Descriptive statistics for group 1 and 2. Note. M—mean; SD—standard deviation.

	Without a Dog				With a Dog			
	Before the Walk (Trial 1)		After the Walk (Trial 2)		Before the Walk (Trial 1)		After the Walk (Trial 2)	
	M	SD	M	SD	M	SD	M	SD
Anxiety State	47.24	11.014	40.94	8.598	46.30	9.631	34.35	6.885
Anxiety trait	55.18	9.869	52.88	10.529	53.90	7.786	48.90	7.166
Fear	2.48	2.551	2.04	2.163	2.57	2.293	1.05	0.973
Resting systolic pressure	125.27	17.271	122.15	16.057	125.05	15.194	126.00	15.437
Resting diastolic pressure	74.42	11.229	74.46	10.550	76.33	12.635	76.43	12.867
Resting pulse	72.46	9.074	71.62	10.269	73.67	13.139	71.05	12.286
Depression	19.69	10.891	-	-	22.00	8.076	-	-
First	26.76	6.078	-	-	24.58	6.678	-	-
Satisfaction after walk	-	-	7.73	1.579	-	-	9.10	1.221

## Data Availability

Data available on request due to restrictions eg privacy or ethical. The data presented in this study are available on request from the corresponding author. The data are not publicly available due to privacy restrictions.

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
