# Peer review of "Can Dog-Assisted Intervention Decrease Anxiety Level and Autonomic Agitation in Patients with Anxiety Disorders?"

_jcm, 2021, doi:10.3390/jcm10215171_

Round 1

Reviewer 1 Report

I commend the authors on addressing the initial comments. They have addressed all of my specific concerns and have done so in a detailed fashion to greatly improve the manuscript. It overall reads more clearly. Great contribution to the literature. 

Author Response

Thank you for the positive assessment of our publication.

Reviewer 2 Report

I enjoyed reading the paper because it presents an interesting project using an animal-assisted intervention, specifically taking a walk with a service dog, for patients with anxiety disorders. With some optimizations, I think the paper would be a contribution to this topic area. The authors have already implemented most of the suggestions, but some of them remained unedited. This is of course fine if there are good reasons to do so. However, in this case, there was a lack of reasoning by the authors for doing so.

The following problems remain:

The relevance of animal-assisted interventions in this field was not discussed (why might AAIs be a promising supplement to existing anxiety therapies?) and the different types are partially inadequately explained/allocated. Still, the study design was neither described nor explained and, in terms of implementation, some questions remain unanswered (e.g., what did the patients do during the time when it was not their turn). Furthermore, the instrument description still lacks more detailed information such as reliability. The description of the analysis procedure and the results are mixed and could be listed separately, as well as divided into physical & psychological variables, which would increase the clarity of the paper. Also missing is a link back to the hypotheses that were made. In the discussion, comparisons and references are inadequate since the variables and samples are different. Also, the correlations and conclusions of the cited studies to the present study are still missing. The conclusion from the results is only partially appropriate because the results of the physical data were only partially significant and were not discussed.

Specific concerns and suggestions

*Abstract: M and SD must be italic, results should be reported after the respective variable (“Among those who walked with a dog, the intervention also led to significant decreases in fear and resting heart rate, F(1, 44) = 11.694, p <0.01, η2 = 0.210 and F(1, 45) = 8.503; p <0.01; η2 = 0.159, respectively”)

*page 1, line 22: Why does it seem so? Here, a reference to existing results in this area that support this assumption would be useful.

*page 1, line 23: I think this sentence would fit better after the sentence "...other forms of therapy are being tested and …".

*page 1, line 33: Again, is “Assistance animal therapy” a subtype of animal-assisted therapy? If so, please mention that. Otherwise: How does it differ from AAT or AAA (goal directed? Trained professional?).

*page 1, line 40: Here, unfortunately, it is unclear to me why this explanation (types of assistance dogs) is necessary for this study.

*page 2, line 1: Here, a lead-in such as "As studies have shown, the presence of a dog..." would be good.

*page 2, line 23: Typo: whther - there is an “e” missing

*page 3, line 29: delete “the”

*page 3, Materials and Methods: Description of the study design is still missing + why/how did you choose this design?

*page 3, line 40: What did the participants do during the mentioned period while it was not their turn?

*page 4, line 14: I think you forgot to mention that they were in the control group (“…from (26 patients (19 female and 7 male) from the control group”.

*page 4, Questionnaires: The description of each questionnaire (source, what they measure, number of items, scaling) should be directly with each questionnaire and also include its Cronbach’s alpha value as well as its source.

*page 5, Results: The heading "Results" is wrong here, because the analysis procedure is also described here. Perhaps a combination of "Analysis and Results" is possible as a heading.

*page 5: There is one word missing in this sentence: “Finally, interaction of the test condition with the resting blood pressure of the subjects in measurement 1 and 2”

*page 5-8: Meaning for your hypotheses (confirmed/rejected)?

*Page 9, line 2: “the” patients - delete “the”

*page 9, line 14ff: In your study, you survey anxiety levels, but then refer to existing literature that shows a reduction in stress from interactions with animals - it would make more sense to refer to literature on anxiety reduction through AAI. In addition, in the subsequent literature references/results, you do not show a connection to your study. Here it would make sense to cite studies that report either the same/similar or differing results on the variables you collected, and then compare them to your results (e.g., improvement of physical measures were found/not found). Furthermore you write that these results are consistent with your hypotheses, however, there was no significant reduction in blood pressure.

*page 10, line 2: Why is this a limitation? If the dog and its owner were in the control group, it would not be a control group.

*page 10, limitations: Another limitation is the small sample size - please also discuss what this means for your results.

*page 10, line 12: AAA may be an effective…

*page 10, line 13: There is a word missing (…modality for the of patients…)

Author Response

Response to Reviewer 2

“*Abstract: M and SD must be italic, results should be reported after the respective variable (“Among those who walked with a dog, the intervention also led to significant decreases in fear and resting heart rate, F(1, 44) = 11.694, p <0.01, η2 = 0.210 and F(1, 45) = 8.503; p <0.01; η2 = 0.159, respectively”)”

Thank you, we have corrected it:

Among those who walked with a dog, the intervention also led to significant decreases in fear F(1, 44) = 11.694, p <0.01, η2 = 0.210 and resting heart rate F(1, 45) = 8.503; p <0.01; η2 = 0.159.”

“*page 1, line 22: Why does it seem so? Here, a reference to existing results in this area that support this assumption would be useful.”

Thank you, we have added 2 references to the studies showing a reduction in anxiety symptoms in patients after the contact with the assistance dog.

“*page 1, line 23: I think this sentence would fit better after the sentence "...other forms of therapy are being tested and …".”

We have moved this sentence to the place pointed by the reviewer.

“*page 1, line 33: Again, is “Assistance animal therapy” a subtype of animal-assisted therapy? If so, please mention that. Otherwise: How does it differ from AAT or AAA (goal directed? Trained professional?).”

We apologize for this obvious error that we overlooked. We misused "Assistance animal therapy" instead of "animal assisted therapy", which was confusing.

AAT and AAA are described in the text:

 “AAT are the professional, goal-directed therapeutic interventions with animals specifically trained for that purpose and are delivered by trained professionals, working formally and directly with healthcare, educators or/and human providers. Animal-Assisted Therapy, especially using horses and dogs, is becoming increasingly common, especially in the treatment of patients with autism spectrum disorders, dementia and PTSD. AAA means interventions that are less structured interactions with animals suitable for this purpose, that have positive impact on the quality of life but are not necessarily delivered by trained professionals.”

“*page 1, line 40: Here, unfortunately, it is unclear to me why this explanation (types of assistance dogs) is necessary for this study.”

It was the suggestion of the first reviewer to includ here information about different types of assistance dogs here.

“*page 2, line 1: Here, a lead-in such as "As studies have shown, the presence of a dog..." would be good.”

Thank you, we have added: “As studies have shown, the presence of a dog may have a positive influence on mood, health and the quality of life…”

“*page 2, line 23: Typo: whther - there is an “e” missing”

Thank you, we have corrected that.

“*page 3, line 29: delete “the””

Thank you, we have delated “the”.

*page 3, Materials and Methods: Description of the study design is still missing + why/how did you choose this design?

Thank you for paying our attention to the unclear design of the study, we have tried to better explain the course of the study. We added information, that patients had psychotherapy in the small subgroups, so they finished they therapy at little bit different time and were ready to take part in our study.

“All participants of our study were recruited from patients treated in this ward with group psychotherapy in little subgroups, that were finishing their therapy between noon and 5 pm.’

and

“All interventions were performed at the same time of the day (between noon and 5 pm), immediately after daily group psychotherapy when all other patients were free to go home. We examined one participant at a time, and, considering the welfare of the service dog, usually 1-3 patients per day (immediately after finishing their therapy).”

We added information why we chose such study design:

We chose such a study design because we wanted to have access to patients with symptomatic anxiety disorders. Patients had contact with dog outside the ward because, due to statute of the hospital, dog wasn’t allowed to enter this building.”

“*page 3, line 40: What did the participants do during the mentioned period while it was not their turn?”

Participants were asked to join the study immediately after finishing their psychotherapy. As psychotherapy for various subgroups was completed in little bit different time,  between noon and 5 p.m., there was no need for the subjects to wait for their turn.

We have added the fragment in parentheses:

“We examined one participant at a time, and, considering the welfare of the service dog, usually 1-3 patients per day (immediately after finishing their therapy).”

“*page 4, line 14: I think you forgot to mention that they were in the control group (“…from (26 patients (19 female and 7 male) from the control group”.”

Thank you very much, we have added “from the control group”.

“*page 4, Questionnaires: The description of each questionnaire (source, what they measure, number of items, scaling) should be directly with each questionnaire and also include its Cronbach’s alpha value as well as its source.”

We have added the descriptions of the scales and counted Cronbach's alpha for scales except for analog scales VAS:

“●         The Beck Depression Inventory (BDI) – 21-item self-report scale, each item consists of responses graded from 0 to 3, to assess the severity of depressive symptoms. [22,23] Cronbach's alpha for our participants was 0.889; performed directly before and after the intervention;

  • The Ford Insomnia Response to Stress Test (FIRST) - a self-rating questionnaire measuring the likelihood of the occurrence of sleep disturbances in response to commonly experienced stressful situations, consists of nine items with evaluation according to a 4-point Likert (1, not likely; 4, very likely), with total score 9-36 [24] Cronbach’s alpha in our study group was 0.859. It was performed directly before and after the intervention;
  • The Brief Symptom Inventory (BSI-18)- a short version of the Symptom-Checklist (SCL-90-R), measures psychiatric symptoms and global symptom load. The BSI-18 comprises 18 items assessing psychological distress in the last seven days on three subscales (Depression, Anxiety, and Somatization) [25]. The Cronbach’s alpha for this scale for our participants was 0.935. It was performed directly before and after the intervention;
  • The visual analogue scale (VAS fear) of level of fear - performed directly before and after the intervention;
  • The State-Trait Anxiety Inventory (STAI) - a questionnaire measuring the level of anxiety and its changes. It consists of 40 items, half of which assess state anxiety, and the other half measure trait anxiety. Participants rate each item on a 4-point scale, ranging from “1=not at all” to “4=totally agree” in case of state anxiety, and “1=almost never” to “4=almost always” in case of trait anxiety. Participants’ responses are summarized, so for each of the two types of anxiety the results range from 20 (very low anxiety level) to 80 (very high anxiety level) [26]. The reliability of the scales in our study was, respectively, for state α =0.932 and trait α =0.875 - performed directly before and after the intervention;”

“*page 5, Results: The heading "Results" is wrong here, because the analysis procedure is also described here. Perhaps a combination of "Analysis and Results" is possible as a heading.”

We have followed the suggestion of the reviewer and rearranged this part of the article. We divided it into two parts – Analytical procedures and Results.

“*page 5: There is one word missing in this sentence: “Finally, interaction of the test condition with the resting blood pressure of the subjects in measurement 1 and 2””

We have added” was done”.

“*page 5-8: Meaning for your hypotheses (confirmed/rejected)?”

We have added: “The obtained results partially confirm our assumptions - contact with the dog was correlated with a decrease in the level of anxiety and a decrease in heart rate, but no significant change in pressure was obtained.”

“*Page 9, line 2: “the” patients - delete “the””

We have deleted “the”.

“*page 9, line 14ff: In your study, you survey anxiety levels, but then refer to existing literature that shows a reduction in stress from interactions with animals - it would make more sense to refer to literature on anxiety reduction through AAI. In addition, in the subsequent literature references/results, you do not show a connection to your study. Here it would make sense to cite studies that report either the same/similar or differing results on the variables you collected, and then compare them to your results (e.g., improvement of physical measures were found/not found). Furthermore, you write that these results are consistent with your hypotheses, however, there was no significant reduction in blood pressure.”

We cited two studies examining anxiety levels after interaction with assisted dogs and using the same STAI scale ( Hoffman, et al., 2009, and Giuliani, et al., 2017):

The STAI scale has been already used before in the studies of the level of anxiety after interactions with the dog [11,12]. In these studies, similarly to our study, a drop in the STAI score was found after contact with the dog. A pilot study by Hoffman et al. (2009) carried out on twelve patients diagnosed with major depression (age: 40.5±10 years) have shown a drop in the STAI state scores after short, 30 minutes contact with the animal. Similarly study by Giuliani et al. (2017) carried out on 53 adults (age: 36.5 ±11.2 years) treated with psychotherapy for learning disabilities confirmed that the company of a dog during psychotherapy resulted in decline in the anxiety score measured with STAI state. Recent study shows the advantage of psychotherapy along with the service dog over psychotherapy itself in veterans with posttraumatic stress disorder (PTSD) in term of depression and the satisfaction of life. The addition of trained service dogs to usual care may confer clinically meaningful improvements in PTSD symptomology [29].”

We have cited studies reporting improvement of physical measures after contact with assisted dogs, and added an attempt to explain why there was no significant drop in blood pressure in our study :

A significant reduction of systolic blood pressure, total cholesterol level and heart rate in the line with significant reduction of cortisol in dog companionship was well proved in the review of cariological studies [33]. However, other trials indicating that AAI could reduce not only heart rate, but also blood pressure [34, 35], in our study we did not detect such a difference. We presume that was because physical activity and curiosity related to meeting the assistance dog were also factors affecting the participants’ blood pressure.”

“*page 10, line 2: Why is this a limitation? If the dog and its owner were in the control group, it would not be a control group.”

It was the first reviewer to suggest adding such a study limitation. We have explained this:

“Another limitation is that the control group walked not only without a dog but also without an owner. So, the study measured an effect of the walk with a team: the owner and the dog on the anxiety level and cardiac parameters”

“*page 10, limitations: Another limitation is the small sample size - please also discuss what this means for your results.”

As suggested, we have added this significant limitation:

“Another limitation is the small sample size thus these results may be incidental and unrepresentative.”

“*page 10, line 12: AAA may be an effective…”

We have corrected this.

*page 10, line 13: There is a word missing (…modality for the of patients…)

We have added “treatment” before “of the patients”.
